# A Retrospective Study of Renal Growth Changes after Proton Beam Therapy for Pediatric Malignant Tumor

**Yinuo Li** [1] **, Masashi Mizumoto** [1,*] **, Yoshiko Oshiro** [1] **, Hazuki Nitta** [1] **, Takashi Saito** [1] **, Takashi Iizumi** [1] **, Chie Kawano** [1] **, Yuni Yamaki** [2] **, Hiroko Fukushima** [2] **, Sho Hosaka** [2] **, Kazushi Maruo** [3] **, Satoshi Kamizawa** [1] **and Hideyuki Sakurai** [1]

1   Department of Radiation Oncology, University of Tsukuba Hospital, Tsukuba 305-8576, Japan
2   Department of Pediatrics, University of Tsukuba Hospital, Tsukuba 305-8576, Japan
3   Department of Biostatistics, Faculty of Medicine, University of Tsukuba, Tsukuba 305-8575, Japan
*   Correspondence: mizumoto@pmrc.tsukuba.ac.jp; Tel.: +81-29-853-7100; Fax: +81-29-853-7102

**Abstract:** The purpose of this study was to analyze renal late effects after proton beam therapy (PBT) for pediatric malignant tumors. A retrospective study was performed in 11 patients under 8 years of age who received PBT between 2013 and 2018. The kidney was exposed in irradiation of the primary lesion in all cases. Kidney volume and contour were measured on CT or MRI. Dose volume was calculated with a treatment-planning system. The median follow-up was 24 months (range, 11–57 months). In irradiated kidneys and control contralateral kidneys, the median volume changes were −5.63 (−20.54 to 7.20) and 5.23 (−2.01 to 16.73) mL/year; and the median % volume changes at 1 year were −8.55% (−47.52 to 15.51%) and 9.53% (−2.13 to 38.78%), respectively. The median relative volume change for irradiated kidneys at 1 year was −16.42% (−52.21 to −4.53%) relative to control kidneys. Kidneys irradiated with doses of 10, 20, 30, 40, and 50 GyE had volume reductions of 0.16%, 0.90%, 1.24%, 2.34%, and 8.2% per irradiated volume, respectively. The larger the irradiated volume, the greater the kidney volume was lost. Volume reduction was much greater in patients aged 4–7 years than in those aged 2–3 years. The results suggest that kidneys exposed to PBT in treatment of pediatric malignant tumor show continuous atrophy in follow-up. The degree of atrophy is increased with a higher radiation dose, greater irradiated volume, and older age. However, with growth and maturation, the contralateral kidney becomes progressively larger and is less affected by radiation.

**Keywords:** pediatric malignant tumor; proton beam therapy; radiation therapy; renal atrophy; late effects

## 1. Introduction

In the multidisciplinary treatment of pediatric cancer, radiotherapy plays an essential role in local control of the tumor [1]. However, patients may experience late effects associated with radiotherapy in various organs [2]. Children who receive chest radiotherapy or whole-body irradiation may be at increased risk for late effects of lung fibrosis, pneumonia, and other diseases [3]. A review in the United States suggested that radiotherapy for pediatric cancer may lead to disorders of the musculoskeletal system, including growth disturbance and skeletal dysplasia, and showed that the risk of growth disturbance is related to the extent and dose of irradiation [4].

The kidney is often irradiated in radiotherapy for pediatric malignant tumors such as childhood neuroblastoma and rhabdomyosarcoma [5,6]. The kidney is a highly radiation-sensitive organ at risk and is susceptible to nephropathy, proteinuria, and hypertension after irradiation [7]. A 2009 study of side effects after radiation in childhood and adolescence indicated that the kidney volumes for exposure to 20 and 30 Gy differed significantly between patients with or without Grade 1 toxicity, with patients with kidney toxicity experiencing higher irradiation exposure to kidney volumes [8].

Proton beam therapy (PBT) has distinct physical characteristics and high accuracy that enables a reduced dosage and irradiated volume of organs at risk, compared to conventional X-ray radiotherapy [9]. Thus, PBT can reduce the incidence and severity of late effects and secondary cancer with an improvement in QoL outcomes in pediatric oncology [10,11], and is increasingly recommended for treatment of pediatric cancer.

To date, late effects after radiotherapy in children and adolescents have been examined mostly in small retrospective studies, and in which only radiation dose can be evaluated due to the shortage of detailed organ dose and volume evaluations [12,13]. Several studies have evaluated late effects for different organs after PBT for pediatric cancer [14–16], but there is still uncertainty regarding long-term follow-up and assessment of renal effects after PBT. No reports have examined renal growth changes after PBT for pediatric malignant tumor. Here, we retrospectively analyzed 11 pediatric patients treated with PBT at our center to investigate the presence and extent of late effects on the kidney, and the association of these effects with radiation dose, volume, and patient age.

## 2. Materials and Methods

### 2.1. Patients

Retrospective study of patients with pediatric malignant tumors who received PBT at our center was approved by the Institutional Review Board. The subjects were 11 patients (5 males, 6 females) aged <8 years at the time of starting PBT between November 2013 and January 2018. The median age at the time of treatment was 3 years (range: 2–7 years). Primary diseases were rhabdomyosarcoma ($n = 2$), neuroblastoma ($n = 8$), and osteosarcoma ($n = 1$).

### 2.2. Proton Beam Therapy

Prior to PBT, all patients underwent a CT scan to allow development of a treatment plan, which was used as a reference for the initial organ morphology. Irradiation was performed daily, once a day on weekdays. All patients were treated with a 155–230 MeV proton beam using the passive diffusion method, shaped with ridge filters, double scattering slices, multiple collimators, and a custom-made bolus to ensure the beam conformed to the planning data. The dose fraction was selected according to the location of the tumor and the organs at risk. The median irradiation dose was 30.6 GyE (range: 19.8 to 70.4 GyE). The median number of fractions was 17 (range: 11–32). The characteristics of the patients, tumors, and PBT are shown in Table 1.

### 2.3. Renal Morphology and DVH Analysis

Long-term follow-up after the end of PBT is essential to assess late effects. In the 11 cases, the median follow-up period was 24 months (range: 11–57 months). The morphology of the kidneys was investigated on pre-treatment CT and on CT or MRI at each follow-up visit after PBT, using MIM Maestro (ver. 6.5.2) software for depicting the contours and calculating the kidney volumes. Dose volume histograms (DVHs) were derived from each patient's treatment plan using VQA ver.2. The % volumes irradiated with 10 GyE (V10), 20 GyE (V20), 30 GyE (V30), 40 GyE (V40), and 50 GyE (V50) were measured. The kidney on the side irradiated at a relatively high dose is referred to as the irradiated kidney, and the kidney on the contralateral side, which was irradiated at a relatively low dose or not irradiated, is regarded as the control. The % changes in volume in the irradiated and control kidneys were investigated.

**Table 1.** Characteristics of the patients (*n* = 11), tumors and PBT.

| Items | *n* | % |
|:---|:---:|:---:|
| Age at start of PBT | | |
|     2 or 3 years | 6 | 54.5% |
|     4–7 years | 5 | 45.5% |
| Gender | | |
|     Male | 5 | 45.5% |
|     Female | 6 | 54.5% |
| Primary disease | | |
|     Rhabdomyosarcoma | 2 | 18.2% |
|     Neuroblastoma | 8 | 72.7% |
|     Osteosarcoma | 1 | 9.1% |
| Total dose and fractions of PBT | | |
|     19.8 GyE/11 fr | 4 | 36.4% |
|     30.6 GyE/17 fr | 4 | 36.4% |
|     41.4 GyE/23 fr | 1 | 9.09% |
|     55.8 GyE/31 fr | 1 | 9.09% |
|     70.4 GyE/32 fr | 1 | 9.09% |
| Chemotherapy | | |
|     Pre-radiation chemotherapy | 8 | 72.7% |
|     Concurrent chemoradiotherapy | 2 | 18.2% |
|     Post-radiation chemotherapy | 1 | 9.09% |
| Median period of follow-up (range) (months) | 24.5 (11–57) | |

*2.4. Statistical Analysis*

Linear mixed-effects models were applied with the % change in kidney volume as the response variable; the binary variable of irradiated (yes/no or less irradiated), the polynomial of the days from irradiation, and their interactions as fixed effects; and the subject as the random effect. The degree of the polynomial was chosen based on the AIC. Marginal means and their confidence intervals were estimated for each level of irradiated (yes/no) at specific time points (24 days and 0.5, 1, 1.5, 2, and 2.5 years). Robust sandwich-variances were used to estimate the confidence intervals. An analysis was also performed that included age groups (2 or 3/≥4 years) and their interactions in the above model. R software ver. 4.2.1 (R Foundation for Statistical Computing, Vienna, Austria) with the lme4 [17], clubSandwich [18], emmeans [19], and AICcmodavg [20] packages were used for calculations. Statistical significance was set at *p* < 0.05.

**3. Results**

*3.1. Treatment Characteristics*

A total of 11 pediatric malignant tumor patients treated with PBT from 2013 to 2018 were included in the study. The patients received chemotherapy before (*n* = 8), during (*n* = 2), or after (*n* = 1) PBT and finally completed their intended PBT treatment (Table 2). Among the 11 patients, three patients (Table 2. #2, #5, #9) were treated for surgery that may have affected the blood supply to the renal vessels of their irradiated kidneys.

**Table 2.** Characteristics of treatment and changes in kidney volume after PBT.

| | Primary Disease | Total Dose (GyE) | Fractions | Chemotherapy | V10 (%) | V20 (%) | V30 (%) | Follow-Up (Month) | Volume Change of Irradiated Kidney/Year (%) | Volume Change of Control Kidney/Year (%) | Relative Volume Change of Irradiated Kidney/Year (%) |
|---|---|---|---|---|---|---|---|---|---|---|---|
| #1 | Rhabdomyosarcoma | 55.8 | 31 | Post-rad | 18.25 | 13.2 | 9.61 | 11 | −8.17 | +12.66 | −18.70 |
| #2 | Neuroblastoma | 19.8 | 11 | Pre-rad | 74.31 | 21.37 | 0 | 37 | −26.41 | +17.87 | −28.73 |
| #3 | Neuroblastoma | 19.8 | 11 | Pre-rad | 54.47 | 15.42 | 0 | 27 | +10.17 | +16.38 | −4.53 |
| #4 | Rhabdomyosarcoma | 41.4 | 23 | Pre-rad | 42.42 | 28.91 | 15.03 | 24 | −10.89 | −2.13 | −9.15 |
| #5 | Neuroblastoma | 30.6 | 17 | Pre-rad | 78.4 | 52.74 | 15.3 | 57 | −2.52 | +9.53 | −8.32 |
| #6 | Osteosarcoma | 70.4 | 32 | CCRT | 32.3 | 25.12 | 15.33 | 28 | −26.38 | +1.04 | −26.78 |
| #7 | Neuroblastoma | 30.6 | 17 | Pre-rad | 99.92 | 98.11 | 34.46 | 16 | −47.52 | +15.08 | −52.21 |
| #8 | Neuroblastoma | 30.6 | 17 | CCRT | 72.12 | 58.52 | 45.49 | 13 | +15.51 | +38.78 | −16.42 |
| #9 | Neuroblastoma | 19.8 | 11 | Pre-rad | 71.84 | 0.47 | 0 | 13 | −8.55 | +5.00 | −12.86 |
| #10 | Neuroblastoma | 30.6 | 17 | Pre-rad | 73.29 | 55.13 | 33.96 | 12 | −34.52 | +0.09 | −34.58 |
| #11 | Neuroblastoma | 19.8 | 11 | Pre-rad | 60.14 | 0.66 | 0 | 42 | −2.68 | +6.98 | −7.77 |

#: Patient number; Pre-rad: Pre-radiation chemotherapy; CCRT: Concurrent chemoradiotherapy; Post-rad: Post-radiation chemotherapy; V10, V20, V30: % volume irradiated by 10, 20, 30 GyE of PBT; +: Trend of volume growth; −: Trend of volume reduction. Relative volume change of irradiated kidneys (%): % change of kidney volume compared to control kidneys.

### 3.2. Renal Volume Changes and DVH Analysis

The median follow-up was 24 months (range 11–57 months). DVH data and the % change in kidney volume after PBT for each patient are shown in Table 2. At the last follow-up, the median kidney volume change for all 11 patients was −5.63 mL/year (−20.54 to 7.20 mL) for irradiated kidneys and 5.23 mL/year (−2.01 to 16.73 mL) for control kidneys. If the kidney volume at the start of irradiation was taken to be 100%, the median % volume changes at 1 year were −8.55% (−47.52 to 15.51%) and 9.53% (−2.13 to 38.78%), respectively. Nine patients had a decrease in volume of the irradiated kidneys. Two patients had an increase in volume (12.84%), but the extent of the increase was much less than that of their control kidneys (27.58%). All 11 patients had a reduction in volume of the irradiated kidneys relative to the controls, with a median relative reduction of 16.42% (4.53 to 52.21%). This indicates that all irradiated kidneys were relatively atrophied during the follow-up period. The control kidneys were enlarged in 10 of the 11 cases. One case had slight shrinkage (2.13%), but the corresponding irradiated kidney had more severe shrinkage (10.89%).

The volume change of the kidneys on both sides in linear mixed-effects model analysis is demonstrated in Figure 1. The degree of the polynomial for days in the model without age was chosen to be 2. All fixed effects and their interactions included in the model were significant. It is evident that the irradiated kidneys (blue line) and control kidneys (red line) followed opposite trends during follow-up. The kidneys exposed to PBT showed significant and continuous atrophy throughout the follow-up period, whereas the control kidneys became progressively larger in size with the growth and maturation of the children, which indicates that they were less affected by radiation.

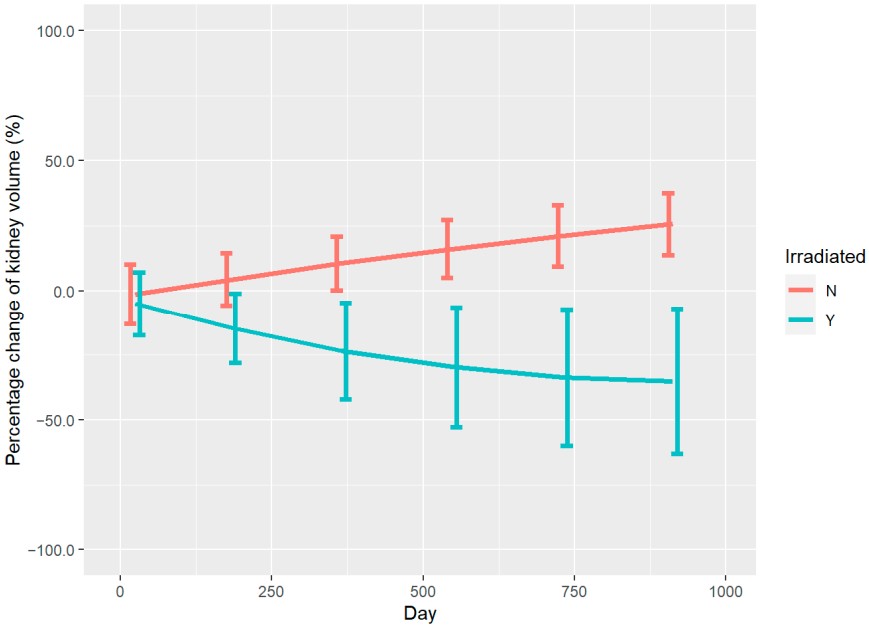

**Figure 1.** Volume changes of the kidneys on both sides during follow-up in mixed effect model analyses. Red line: control kidneys. Blue line: irradiated kidneys.

The relative atrophy rate per year at the corresponding dose of irradiation is shown in Figure 2. Based on the numbers on the graph (A, B), if B, which is the irradiated volume percentage of kidneys, is larger under the same irradiation dose, the relative atrophy rate increases, especially at 20 GyE and 30 GyE. Kidneys irradiated with doses of 10, 20, 30, 40, and 50 GyE had median relative reduction rates of 10.32%, 16.63%, 34.58%, 12.79%, and 22.74% per year, respectively, which corresponded to reduction rates of 0.16%, 0.90%, 1.24%, 2.34%, and 8.2% per unit of irradiated volume, respectively. This result suggests

that the atrophy extent per irradiated volume may be more significant when the kidney is irradiated with the higher irradiation doses.

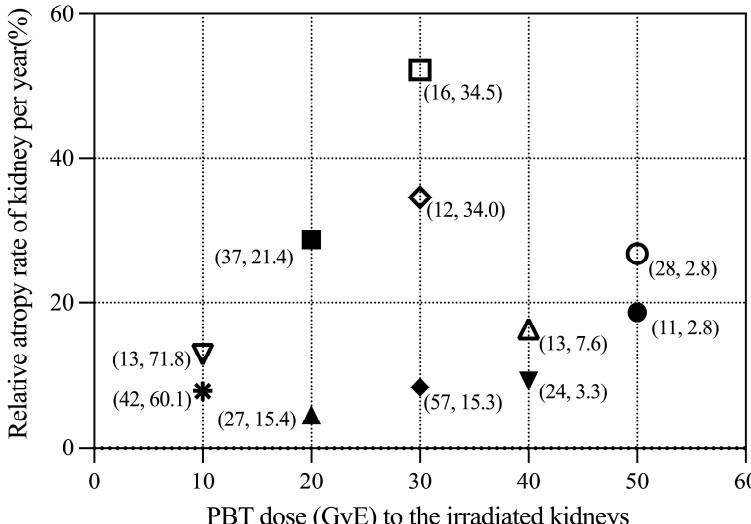

**Figure 2.** Relative atrophy rate of irradiated kidneys per year with irradiation of different doses of PBT. The different symbols represent information about each patient. Relative atrophy rate is that compared with the unirradiated or lower dose-irradiated control contralateral kidney. Numbers in (): the former is the follow-up time (months) for each patient and the latter is the % volume of the kidney irradiated by the corresponding dose.

A typical case (#2 in Table 2) showing morphological changes of the bilateral kidneys after PBT is illustrated in Figure 3. The patient was a 4-year-old neuroblastoma survivor with a primary lesion in the left adrenal glands. After chemotherapy and surgery to remove the tumor, the patient was referred to our hospital for PBT. Irradiation of 19.8 GyE in 11 fractions of PBT was administered once daily. CT before PBT showed no significant difference in the volume of the two kidneys (Figure 3a). The dose distribution diagrams in the cross-sectional and coronal planes (Figure 3b,c) show that the left kidney was extensively covered by PBT, whereas the right kidney was barely involved. MRI on day 52 after completion of PBT (Figure 3d) showed that the left kidney had begun to show a tendency to reduce in size, while the right kidney has a slightly enlarged indication. One year after irradiation, the left kidney had shrunk markedly, but the right kidney had increased in size with growth and development (Figure 3e). At the last follow-up visit 3 years after irradiation, MRI (Figure 3f) revealed further atrophy of the left kidney, whereas the right kidney had enlarged progressively, giving a significant volume difference between the left and right sides.

As shown in Figure 4, there were differences in the % change of kidney volume over time after irradiation in children in different age groups. The degree of the polynomial for days in the model including age was chosen to be 1. All fixed effects and their interactions included in the model were significant. Children aged 4–7 years (dashed line) had a significant reduction in volume of the irradiated kidneys (dashed blue line). However, children aged 2 or 3 years (solid blue line) had a smaller decrease in volume, although there was still an overall trend for shrinkage. Control kidneys (red line) in both age groups showed a similar tendency to increase in volume and the difference in the degree of enlargement was not significant.

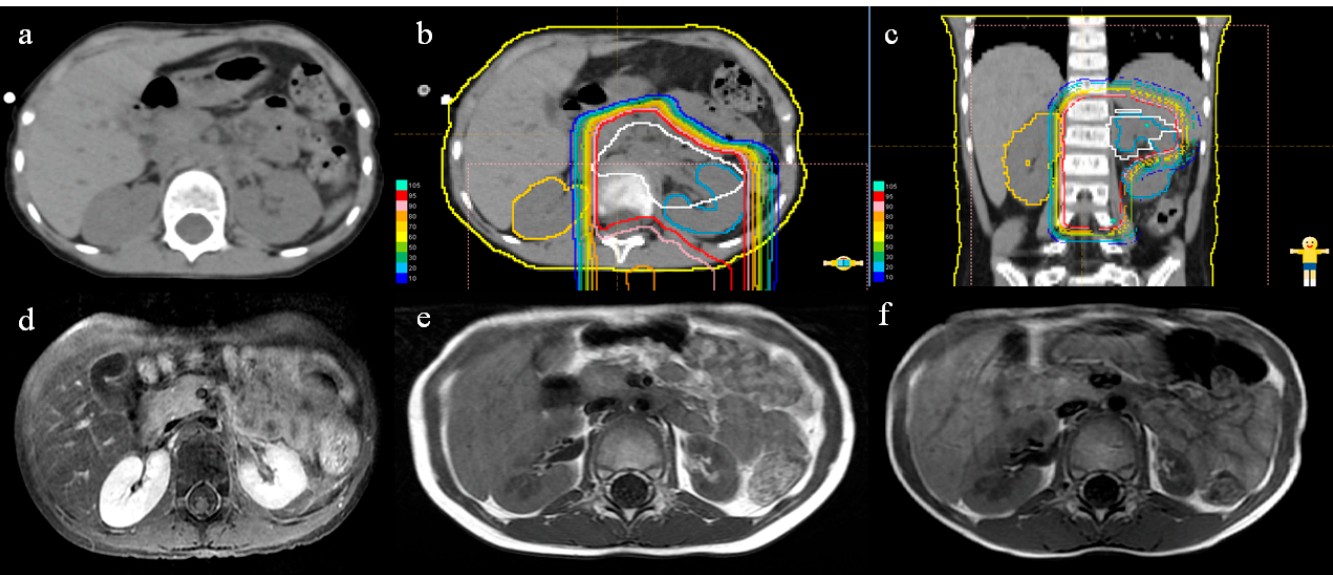

**Figure 3.** Abdominal CT and MRI before and after PBT and dose distribution diagrams. (**a**) CT before PBT. (**b**) Dose distribution of PBT in cross-section. (**c**) Dose distribution of PBT in the coronal plane. (**d**–**f**) MRI at 52 days (**d**), 1 year (**e**), and 3 years (**f**) after completion of PBT.

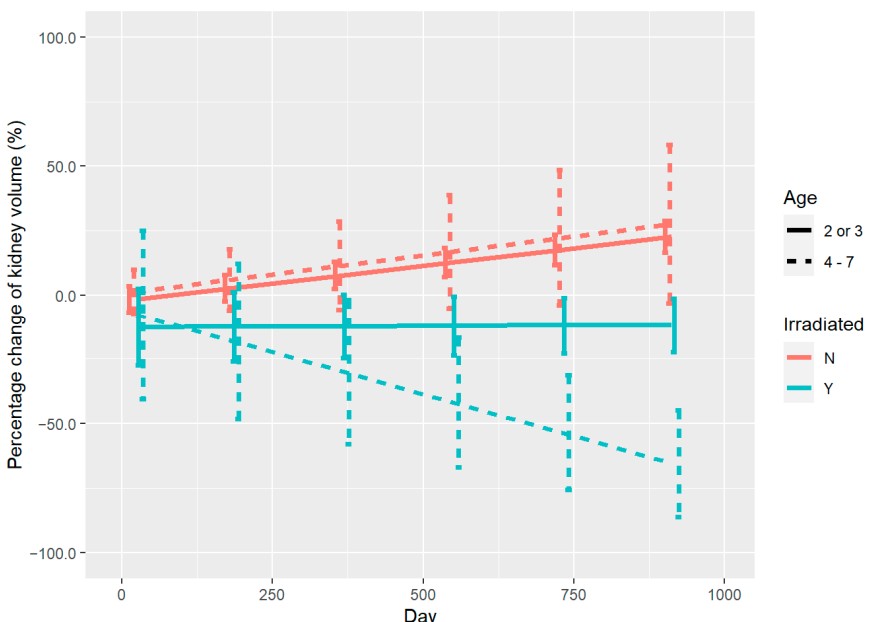

**Figure 4.** % change of kidney volume during follow-up in children of different ages. Red line: control kidneys. Blue line: irradiated kidneys. Dashed line: 4–7 years old. Solid line: 2 or 3 years old.

### 3.3. Analysis of Renal Function Parameters

Before PBT, the median serum creatinine (s-Cr) level in the 11 patients was 0.29 (0.20–0.45) mg/dL and the median glomerular filtration rate (eGFR) was 116.72 (74.6–196.81) mL/min/1.73 m$^2$. At the end of PBT, median s-Cr was 0.29 (0.19–0.42) mg/dL and eGFR was 123.51 (84.99–207.8) mL/min/1.73 m$^2$. Based on the classification of Japanese pediatric CKD, most patients presented with renal function in stage 1, and the differences in s-Cr and eGFR before and after PBT were not significant. Three patients (#1, #5, and #8) were followed up for more than 6 months after completion of PBT. For these patients, s-Cr levels at the start of PBT and at the last follow-up were 0.31/0.25/0.38 mg/dL and 0.31/0.43/0.29 mg/dL, respectively, and all were within the normal range.

## 4. Discussion

More than 80% of children treated for cancer now survive for more than 5 years because of advances in therapeutic approaches [21]. However, the treatments that promote survival from cancer may also cause health problems in later life [22–24]. Radiotherapy utilizes low and high linear energy transfer (LET) rays to kill tumor cells, but these rays also cause damage to normal tissue [25]. A study at the M.D. Anderson Cancer Center of pediatric patients with non-surgical treatment and follow-up revealed severe post-treatment side effects in the irradiated area, including jaw, orbital, or hemifacial dysplasia with varying degrees of overlying soft tissue atrophy [26]. In a study of 42 patients with pediatric Wilms' tumor, the rates of muscle dysplasia, limb length inequality, kyphosis and iliac wing dysplasia after radiotherapy were 16.7%, 11.9%, 7.1%, and 7.1%, respectively. Scoliosis appeared in 42.9% of cases and late effects of treatment were observed in more than two-thirds of the patients [27].

The number of cases of pediatric tumors treated with PBT is relatively small and details of outcomes are lacking, but the radiation dose and volume distribution based on photon baseline data and clinical experience suggest that PBT has an advantage of reducing or eliminating many early and late effects of radiotherapy [28,29]. In Japan, PBT is included in health insurance for pediatric cancer treatment as advanced radiotherapy [11]. Previously, our center has investigated the effects of radiotherapy and PBT on vertebral growth in the spine of children. The results showed that the growth rate of vertebrae decreased with an increasing irradiation dose, regardless of the treatment modality of X-ray or PBT [30–32]. Mizumoto et al. found that the reduced radiation dose to normal organs in PBT may lower the risk of late adverse events, and a reduced risk of growth restriction, endocrine dysfunction, reduced fertility, and secondary cancers was shown in children and adolescents [15].

Radiation-induced kidney toxicity is a major concern for radiation oncologists, which may present as acute nephropathy in as soon as 6 months, such as chronic nephropathy, hypertension, or asymptomatic proteinuria [33]. Most studies of the late effects of radiation nephropathy have been conducted in children with Wilms' tumor and have generally shown that children receiving higher radiation doses are at higher risk of developing renal insufficiency [34–36]. QUANTEC data show that up to 50% of patients exposed to an average dose of >18 Gy to both kidneys in X-ray radiotherapy develop clinically relevant renal dysfunction [37]. However, Fukumitsu et al. found that high doses of PBT could control renal cell carcinoma while maintaining renal function, with no adverse events of grade 3 or higher and no significant effects on serum urea nitrogen and creatinine levels [38].

In the current study, 11 pediatric patients aged 2–7 years old treated with PBT had a median follow-up period of 24 months (range 11–57 months). At the final follow-up, the irradiated kidneys of all 11 patients exhibited relative atrophy compared with control contralateral kidneys, regardless of the irradiation dose, suggesting that the kidneys not only respond to X-rays, but also to proton beams. Ten patients had a volume increase in control kidneys due to growth and maturation, which indicates that PBT has only a slight effect on development of the contralateral kidney. This reflects the advantage of the accuracy of PBT based on its physical characteristics. The degree of renal atrophy was related to the radiation dose, irradiated volume, and patient age. Within a unit of irradiated volume, a distinct degree of atrophy was observed at high doses. Moreover, at a certain irradiation dose, the atrophy rate increased as the irradiated volume became larger. Renal atrophy was more significant in older children (aged 4–7 years). This may be because it is easier to observe volume changes when the kidneys start to grow in patients who are a little older. We also found a relatively high rate of renal atrophy even with irradiation at lower doses (≤20 GyE), but with a larger irradiation volume. This suggests that the region of exposure for pediatric patients requires more careful planning.

Renal volume is a useful parameter for assessment of renal function. Several studies have also reported a correlation between GFR and renal volume. For example, Gong

et al. found that kidney volume was strongly correlated with CG GFR (r = 0.615) and less correlated with MDRD GFR (r = 0.432) [39], and Shin et al. showed that total renal volume was significantly correlated with CG GFR (r = 0.43) in 113 healthy young men [40]. However, in our study, although kidney volume was reduced after PBT in all cases, no cases developed renal insufficiency during follow-up. This indicates that PBT can treat the primary disease while maintaining renal function.

In a study of normal kidney volume changes in growth and development of 321 Asian children, Rongviriyapanich et al. found mean growth rates for the left and right kidneys of 11.48% and 7.66% per year, respectively, and mean increases in kidney volume of 8.60% per year for males and 11.45% per year for females [41]. In the control kidneys in the current study, the mean volume increases per year were 10.42% on the left side (*n* = 8) and 12.65% on the right side (*n* = 3), and males and females had mean increases of 9.91% and 11.96% per year, respectively. These data show that the enlargement of the non-irradiated control kidneys in our patients was generally consistent with normal growth, and that there was little evidence of compensatory enlargement. These data further show that PBT has little effect on growth of the kidney in the non-irradiated field.

Because patients surviving for 5 years are still at higher risk of progression to subsequent malignancy, chronic disease, and dysfunction, such as radiation nephropathy or chronic renal insufficiency [42,43], and this makes it important to monitor these patients over a longer follow-up period. The current study was limited by the small number of cases and irradiation sites. However, the results do indicate a positive effect of PBT on pediatric malignant tumors, without major adverse effects on renal function. A further study in more patients is needed to confirm the beneficial effects of PBT for these tumors.

## 5. Conclusions

After proton beam therapy for pediatric malignant tumor, kidneys exposed to irradiation exhibited significant atrophy. The degree of atrophy correlated with the radiation dose, irradiated volume, and patient age. In contrast, the contralateral kidneys showed a normal increase in volume with growth of the children and were unaffected by irradiation.

**Author Contributions:** Conceptualization, M.M. and Y.O.; methodology, M.M.; software, K.M.; validation, C.K., T.I., S.H., K.M. and S.K.; formal analysis, K.M.; investigation, Y.L., H.N., T.S., Y.Y., H.F. and H.S.; resources, H.S.; data curation, Y.L.; writing—original draft preparation, Y.L.; writing—review and editing, M.M.; visualization, Y.O.; supervision, M.M.; project administration, H.S.; funding acquisition, H.S. All authors have read and agreed to the published version of the manuscript.

**Funding:** This research was supported in part by Mitsui Life Social Welfare Foundation.

**Institutional Review Board Statement:** The study was conducted in accordance with the Declaration of Helsinki and approved by the Institutional Review Board (or Ethics Committee) of Tsukuba Clinical Research & Development Organization (H30-099, 14 November 2019).

**Informed Consent Statement:** Informed consent was obtained from the parents of all subjects involved in the study.

**Data Availability Statement:** Data are available from the corresponding author upon request.

**Conflicts of Interest:** The authors declare no conflict of interest. The funders had no role in the design of the study; in the collection, analyses, or interpretation of data; in the writing of the manuscript; or in the decision to publish the results.

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
