# Peer review of "A Retrospective Study of Renal Growth Changes after Proton Beam Therapy for Pediatric Malignant Tumor"

_curroncol, doi:10.3390/curroncol30020120_

Round 1

Reviewer 1 Report

The authors investigated renal late effects after proton beam therapy for pediatric malignant tumors. The study was performed in 11 patients aged 2-7 years old of age in 2013 to 2018. The manuscript is well prepared, the results and the experiment, and all figures are properly addressed.

Author Response

Response to Reviewer 1 Comments

Point 1: The authors investigated renal late effects after proton beam therapy for pediatric malignant tumors. The study was performed in 11 patients aged 2-7 years old of age in 2013 to 2018. The manuscript is well prepared, the results and the experiment, and all figures are properly addressed.

Response 1: Thank you for your review and positive comments.

Reviewer 2 Report

Dear authors,
This is a very interesting article.
I would just like to ask if there are markers of renal function in these children
at follow-up (eg creatinine, cystatin-c, egfr). I think they should be reported.
Also, was the increase in the contralateral kidney attributed to compensatory
hypertrophy or is it within the curve for age amd gender?
Also i think that you have to improve ypur conclusion paragraph.
Please add author contrbutions.

Author Response

Response to Reviewer 2 Comments

Point 1: Dear authors,

This is a very interesting article. I would just like to ask if there are markers of renal function in these children at follow-up (eg creatinine, cystatin-c, egfr). I think they should be reported

Response 1: Thank you for your careful reading and helpful comments. As the reviewer suggests, we have added a paragraph in the Results (Page 7, Line 198-207) on analysis of the levels of serum creatinine and eGFR before and after irradiation in all cases. Long-term follow-up data were also obtained in three cases.

Point 2: Also, was the increase in the contralateral kidney attributed to compensatory hypertrophy or is it within the curve for age and gender? Also I think that you have to improve your conclusion paragraph.

Response 2: Thank you for this suggestion. We checked the literature and we found that the volume increase of the unirradiated kidneys in this study is similar to that of normal growth in healthy children in terms of age and gender. We have added this information in the Discussion (Page 8, Line 269-278). In addition, we have refined and improved the Conclusion (Page 8, Line 288-292).

Point 3: Please add author contributions.

Response 3: The author contributions have been listed below the Conclusion (Page 9, Line 294-300).

Reviewer 3 Report

The strongest limitation of the study is a small number of its participants. Moreover, I miss information on kidney parameters before and after PBT in presented children. Although the authors mentioned about renal function of the study patients in the discussion (none patients presented with renal insufficiency in the follow up) those data should be analyzed and included also in the material and results section.

The statement “…., there was still a suspected residual tumor in the lymph nodes of the left hilum” should be revised.

The English editing is needed.

In my opinion presented manuscript shows important issue and should be published after major revision.

Author Response

Response to Reviewer 3 Comments

Point 1: The strongest limitation of the study is a small number of its participants. Moreover, I miss information on kidney parameters before and after PBT in presented children. Although the authors mentioned about renal function of the study patients in the discussion (none patients presented with renal insufficiency in the follow up) those data should be analyzed and included also in the material and results section.

Response 1: Thank you for your careful reading and helpful comments. As the reviewer suggests, we have added a paragraph in the Results (Page 7, Line 198-207) on analysis of the levels of serum creatinine and eGFR before and after irradiation in all cases. Long-term follow-up data were also obtained in three cases.

Point 2: The statement "...., there was still a suspected residual tumor in the lymph nodes of the left hilum" should be revised. The English editing is needed.

Response 2: We decided to delete this part of the sentence. English proofreading has also been performed by a native speaker.

Point 3: In my opinion presented manuscript shows important issue and should be published after major revision.

Response 3: Thank you for your review of the paper.

Round 2

Reviewer 3 Report

Thank you for corrections and additions made.